# Water Resistance of Super Adhesive Emulsified Asphalt Based on Dynamic Water Scouring

**Xiujun Li [1], Yue Sun [1], Ningning Wang [2], Fangzhi Shi [3] and Bin Peng [1,\*]**

[1]   School of Environment and Architecture, University of Shanghai for Science and Technology, Shanghai 200093, China; junzixiu@usst.edu.cn (X.L.); 212272035@st.usst.edu.cn (Y.S.)
[2]   Shanghai Municipal Engineering Design Co., Ltd., Shanghai 200093, China; nndaling@163.com
[3]   Road Engineering Technology Research Institute Co., Ltd., Jiaxing 314000, China; alan.shi@wirtgen-group.com
[\*]   Correspondence: binpeng@usst.edu.cn

**Abstract:** This study aimed to address the issue of inadequate water resistance in asphalt pavement tack coat materials by preparing super adhesive emulsified asphalt (SAE). Due to the lack of consideration of dynamic water scouring in previous studies on the water resistance analysis of tack coat materials, this research employed self-designed inclined shear, pullout, and dynamic water scouring devices to examine the shear and tensile strength of matrix emulsified asphalt, styrene block copolymer (SBS)-modified emulsified asphalt, and SAE under various conditions of sprinkling volumes, temperatures, and scouring times. The findings indicate that, in the dynamic water scouring test conducted under the optimal sprinkling amount of tack coat material and the most unfavorable temperature conditions between layers, SAE exhibited residual shear and tensile strengths that were higher than SBS-modified emulsified asphalt and matrix emulsified asphalt. Combining polymerized styrene butadiene rubber (SBR) and polyurethane (PU) as two modifiers in the mix exhibits a synergistic effect, enhancing the bonding performance, scouring resistance, and water resistance of SAE as the tack coat material.

**Keywords:** modified emulsified asphalt tack coat; residual shear strength ratio; residual tensile strength ratio; dynamic water scouring; water resistance

## 1. Introduction

In the context of the transition toward heavier and multi-axle traffic loads, coupled with the influence of surrounding environmental factors, higher demands have been placed on the mechanical performance of asphalt pavement layers. Generally, the surface layer structure of the pavement is divided into two or three sub-layers. After compaction in layers, the surface of the structural layer becomes relatively smooth, resulting in a significant reduction in interlayer frictional resistance. The tack coat in asphalt pavement plays a crucial role in enhancing the bond between layers and coordinating deformation. Insufficient interlayer bonding performance in the pavement structure can result in interlayer sliding under significant vertical and horizontal loads, leading to issues such as nudging, congestion, and other problems that negatively impact pavement performance and reduce lifespan [1–3].

In the current practice, emulsified asphalt is commonly used as a tack coat material. However, with the gradual development of the transportation industry, unmodified emulsified asphalt cannot meet the requirements for road use. Researchers mostly employ high-molecular-weight polymers to modify emulsified asphalt, thereby further improving its road performance. Currently, the main modifiers used for emulsified asphalt include styrene butadiene rubber (SBR), styrene butadiene styrene block copolymer (SBS), and waterborne epoxy resin (WER). Jada et al. [4] introduced SBS into the base emulsified asphalt and observed alterations in the surface charge of the base emulsified asphalt, which affected

the emulsion's performance. Abedini et al. [5] utilized SBR latex for the modification of emulsified asphalt and concluded that SBR could enhance the low-temperature flexibility and ductility of asphalt. Mcnally [6] prepared polyurethane-modified asphalt and found that, under the same dosage conditions, the viscosity of polyurethane-modified asphalt was significantly higher than that of SBS-modified asphalt. Poulikakos [7] summarized that the recycling of high-molecular-weight materials such as polyurethane, compared to traditional asphalt modifiers, could reduce carbon dioxide emissions and lower construction costs. Bazmara [8] conducted testing and analysis of polyurethane-modified asphalt using Fourier-transform infrared spectroscopy (FTIR), and the results indicated that the free isocyanate groups in polyurethane could react with hydroxyl groups in asphalt. Zhang [9] et al. developed a high-performance waterborne polyurethane-modified emulsified asphalt. Their research demonstrated that the addition of a waterborne polyurethane modifier significantly improved the high-temperature stability of emulsified asphalt.

Polyurethane (PU), as a rapidly emerging organic high-molecular-weight material, performs better in modifying asphalt in terms of durability and elastic recovery capabilities compared to traditional polymer modifiers. However, the introduction of resinous materials can reduce the low-temperature flexibility of asphalt. Given that SBR possesses excellent high- and low-temperature characteristics, particularly low-temperature flexibility, this study employs PU and SBR in combination to develop SAE tack coat materials by using composite modification techniques and investigates the key preparation parameters of composite-modified emulsified asphalt.

The current analysis of interlayer performance commonly involves shear, pullout, and water immersion tests [10–13]. However, there is still insufficient research on water resistance. For example, Li [14] conducted a comparative analysis of the impact of moisture on the interlayer bond strength through water immersion regeneration. Wang [15] assessed the penetration effect and waterproof performance of two types of tack coat materials, namely waterborne epoxy resin emulsified asphalt and single emulsified asphalt, using penetration depth and water penetration tests. Additionally, Liu [16] employed a transverse water penetration test in combination with digital image technology to evaluate the degree of water penetration and accumulation in the tack coat, thereby quantifying the distribution of voids within the tack coat. In conclusion, the existing analysis of water resistance in tack coat materials primarily concentrates on static water immersion and water penetration tests. However, these research approaches neglect the real-world conditions of asphalt pavement, including the impact of surface water and dynamic water scouring caused by vehicular loads.

The road surface is prone to water film formation due to various factors such as precipitation, rising water table, and temperature variations. Moreover, the issue of vehicle overload and heavy loads exacerbates road conditions, leading to the formation of numerous micro-cracks on the upper layer of the asphalt pavement [17]. The analysis of the causes of dynamic water pressure reveals that each vehicle passing over the road surface generates a "pumping" effect, as illustrated in Figure 1. During this process, water is forced into the adhesive layer through micro-cracks on the road surface, and the water under dynamic water pressure exerts a scouring effect on the tack coat [18–21]. Consequently, the bond strength of the tack coat gradually deteriorates over time due to the combined effects of temperature and dynamic water pressure. Adequate water resistance in the tack coat material can mitigate its shedding and interlayer slippage damage to a certain extent.

Given the preceding analysis, there is a crucial need to investigate the waterproofing and impermeability of tack coat materials. Thus, this paper examines the shear and tensile strengths of the tack coat by conducting tests under various conditions, including different treatment materials, sprinkling volumes, temperatures, and dynamic water flushing durations. These tests were performed using self-designed inclined shear, pullout, and dynamic water scouring devices. Additionally, the ratios of residual shear and tensile strength ratios following dynamic water scouring were determined to evaluate the water resistance performance of SAE and analyze the underlying water resistance mechanism.

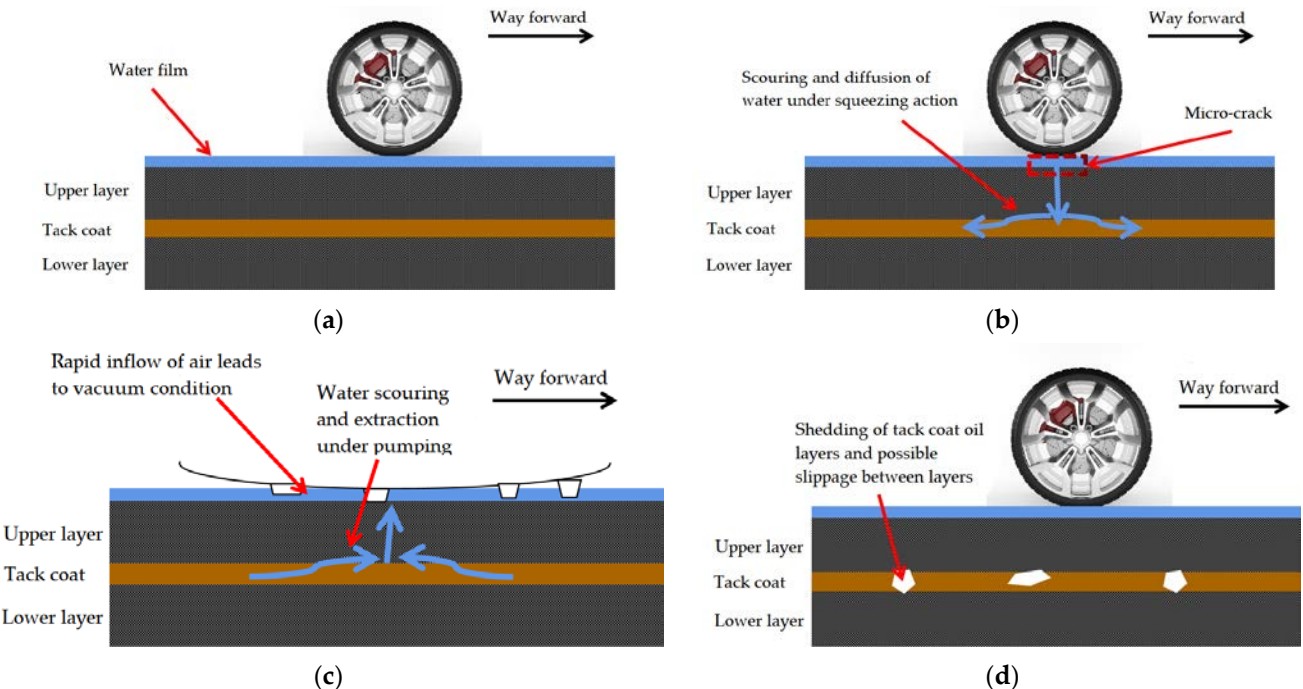

**Figure 1.** Diagram of dynamic water scouring simulation: (**a**) Simulation of car tire driving process; (**b**) Water scouring caused by tire compression; (**c**) Water scouring caused by tires driving away; (**d**) Part of the tack coat oil comes off after dynamic water flushing.

## 2. Preparation of SAE

### 2.1. Raw Materials

The matrix asphalt was produced by Sinopec Group. The main technical indexes are shown in Table 1. The waterborne polyurethanes were bluish, translucent liquid with a solid content of 40%. The styrene butadiene rubber (SBR)-modified latex was cationic and had a solid content of 50%. The emulsifier used a cationic fast-cracking type emulsifier. The matrix emulsified asphalt and SBS-modified emulsified asphalt were supported by a company in Jiaxing City, Zhejiang Province.

**Table 1.** The main technical indexes of matrix asphalt.

| Item | Test Value | Specification | Test Method |
|---|---|---|---|
| Penetration (25 °C, 100 g, and 5 s)/(0.1 mm) | 67.4 | 40~120 | T0604 |
| 15 °C Ductility (5 mm/min)/cm | 112 | ≥100 | T0605 |
| Softening point/°C | 52 | ≥50 | T0606 |

### 2.2. Preparation of Emulsified Asphalt

A colloid mill prepared the cationic emulsified asphalt, the selected emulsifier was mixed with a 2% concentration, and the oil-water ratio was 6:4. The concentrations of SBR modifier latex were chosen to be 2%, 4%, and 6% [22], and the concentrations of PU modifier were 10%, 12%, and 14%. The key factors to consider include the dosage of the two modifiers and the blending rate during the modification phase. The optimal preparation scheme of SAE was determined using the orthogonal test method, with 4% SBR and 12% PU. The mixture was stirred at 500 rotations per minute for 30 s to ensure thorough blending. Then, in the mentioned solution, a specific amount of emulsified asphalt was added and stirred at a high stirring speed of 500 revolutions per minute for 10 min [9,23]. The prepared SAE, matrix emulsified asphalt, and SBS-modified emulsified asphalt underwent conventional performance tests. The test results are shown in Table 2.

**Table 2.** The main technical indexes of emulsified asphalt.

| Item | | Matrix Emulsified Asphalt | SBS-Modified Emulsified Asphalt | SAE |
|---|---|---|---|---|
| Particle charge | | Cation | Cation | Cation |
| Retained on 1.18 mm sieve | | 0.09 | 0.04 | 0.02 |
| Evaporation residue content | | 60 | 52 | 55 |
| Evaporation residues | Penetration/(0.1 mm) | 78 | 51 | 42 |
| | 5 °C Ductility/cm | 38 | 50 | 61 |
| | Softening point/°C | 42 | 55 | 76.4 |

## 3. Test Methods

To study the performance of emulsified asphalt tack coat oil, it is necessary to simulate the pavement structure for the adhesive layer test. The chosen structure, as shown in Figure 2, consists of an "AC-13 (Asphalt concrete with a maximum nominal grain size of not more than 13 mm) asphalt surface layer, adhesive layers, and AC-20 (Asphalt concrete with a maximum nominal grain size of not more than 20 mm) asphalt surface layers". To reduce the mixture volume and simplify the complex process of core drilling and sampling, the height of the upper and lower layers of the test specimen was set to half the height of the Marshall specimen. Subsequently, the calculated amount of matrix emulsified asphalt, SBS-modified emulsified asphalt, and SAE was evenly applied to the surface of AC-20 test pieces. The coated pieces were then placed flat in a clean, dry place. AC-13 test pieces were added to create a composite test piece for inclined shear and pullout testing. The gradation of this composite specimen is provided in Table 3.

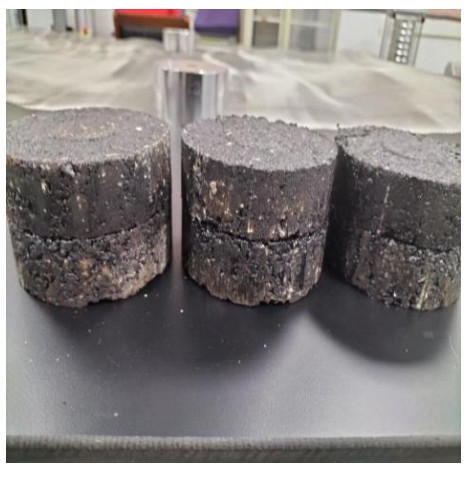
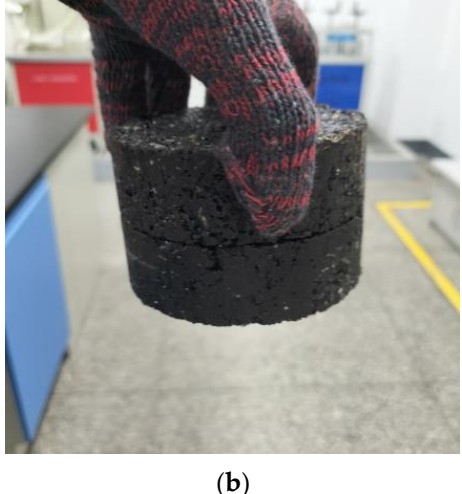

(**a**)  (**b**)

**Figure 2.** The preparation of composite specimen: (**a**) The preparation of composite specimen; (**b**) Composite specimen after forming.

**Table 3.** The gradation of the mixture.

| Mixture Type | Sieve Size/mm | | | | | | | | | | | |
|---|---|---|---|---|---|---|---|---|---|---|---|---|
| | 26.5 | 19.0 | 16.0 | 13.2 | 9.5 | 4.75 | 2.36 | 1.18 | 0.6 | 0.3 | 0.15 | 0.075 |
| AC-13 | 100.0 | 100.0 | 100.0 | 96.8 | 82.8 | 56.2 | 31.4 | 24.0 | 16.4 | 10.0 | 6.0 | 5.0 |
| AC-20 | 100.0 | 86.5 | 66.0 | 45.3 | 27.0 | 17.4 | 11.4 | 8.4 | 6.4 | 4.9 | 4.1 | 3.5 |

### 3.1. Dynamic Water Scouring Test

#### 3.1.1. Dynamic Water Scouring Simulation Test Device

This paper employed an independently designed and processed dynamic water scouring device for experimental purposes. The schematic diagram of the dynamic water-scouring simulation device is depicted in Figure 3. Du [24] derived the conclusion that, through simulation, the action time of pore water pressure generated by a car driving over the road is 0.1 s when the peak pore water pressure is 0.2 MPa. Therefore, this test utilized a pressure control device to maintain a constant air pressure of 0.2 MPa in the pressure vessel. The water-saturated specimen was then pressurized for 10 min, 20 min, and 30 min, respectively, to simulate the scouring effect equivalent to 6000, 12,000, and 18,000 times the pore water pressure generated by a car driving through. Following the scouring process, the specimens were promptly removed and subjected to shear and tensile strength testing following the inclined shear and pullout test procedures. The residual shear and tensile strength ratios from dynamic water scouring were calculated. The water used in the dynamic water souring test was ordinary domestic water, which was used to simulate the effect of dynamic water scouring on the tack coat after natural rainfall.

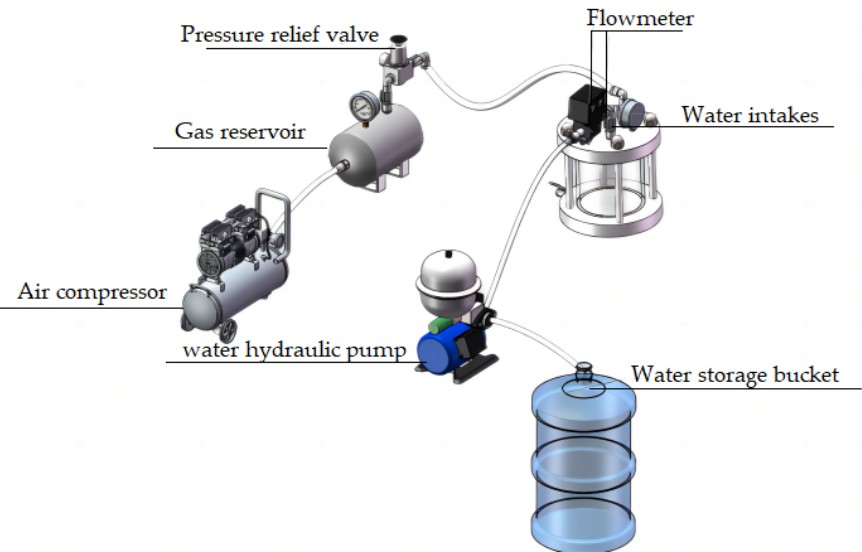

**Figure 3.** Diagram of dynamic water scouring simulation device.

#### 3.1.2. Test Temperature Determination

Temperature significantly influenced the various performance aspects of the viscous layer oil. However, the ambient, road table, and interlayer temperatures exhibited noticeable inconsistencies. Consequently, this study determined the interlayer temperature for the experiment by utilizing the strategic highway research program's (SHRP) temperature prediction model [25,26]. The regression model for the interlayer temperature is presented in the following equation:

$$T_{d(max)} = \left(T_{s(max)} + 17.78\right)\left(1 - 2.48 \times 10^{-3}d + 1.085 \times 10^{-5}d^2 - 2.441 \times 10^{-8}d^3\right) - 17.78 \tag{1}$$

$$T_{d(min)} = T_{s(min)} + 5.1 \times 10^{-2}d - 6.3 \times 10^{-5}d^2 \tag{2}$$

where $T_{d(max)}$ is the maximum temperature at a specific depth of the road table, °C; $T_{s(max)}$ is the maximum temperature of the road table, °C; $T_{s(min)}$ is the minimum temperature of the road table, °C; $d$ is the depth of the oil-sticking layer from the road surface, mm.

Based on the measurements, it is shown that the maximum temperature of the asphalt pavement at the project site can reach 60 °C, while in winter, the minimum temperature of the asphalt pavement can drop to 3 °C. The construction process involved spraying tack coat oil after milling a portion of the old pavement, followed by adding a 4 cm layer of new pavement ($d$ = 40 mm). Calculations based on the above formula show that the highest and

lowest temperatures between the layers were 53.5 °C and 4.9 °C respectively. Therefore, the most unfavorable temperatures selected for the test were 55 °C and 5 °C, respectively.

*3.2. Interlayer Shear Test*

The interlayer performance of asphalt pavements has been investigated mainly by direct shear and inclined shear tests. Compared with the direct shear test, the inclined shear test can produce normal stress by designing the loading mold, which can simulate the actual stress state of pavements to a greater extent. Previous research has shown the differences in the shear angles used in the inclined shear tests. Hu [27] chose a 40° inclined shear test based on the shear angle calculated by the finite element method of Tongji University. Mou [28] used a self-made 45° inclined shear fixture to study the shear strength between epoxy asphalt ultra-thin overlay and conventional asphalt pavement. The interlayer shear stress is greatly affected by the shear angle, which can be determined according to the traffic design load, traffic volume, actual traffic load, etc., of the studied pavement. Commonly, the shear angle is 32° for road sections without longitudinal slope, up to 37° for road sections with frequent speed changes, and close to 45° for special road sections (longitudinal slope greater than 5%). Therefore, many scholars consider the most unfavorable angle, choosing 45° as the shear angle in the test [29].

Consequently, considering the most unfavorable angle, a 45° inclined shear test was selected in the current study. The shear instrument used in this paper is a combination of self-designed molds and a universal testing machine based on the DLG-A type pavement shear tester developed by Wu et al. [30]. The inclined shear mold is shown in Figure 4. Considering the short contact time between the tire and the road during the actual driving process of a car, a loading rate of 50 mm/min was used for fast loading [31].

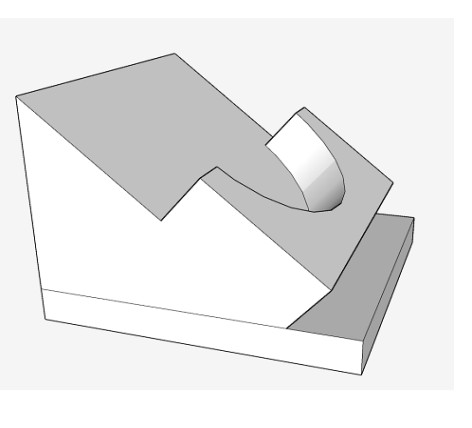

(**a**)

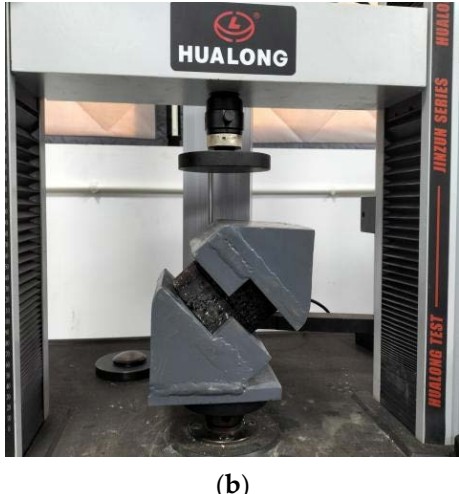

(**b**)

**Figure 4.** Inclined shear test: (**a**) Diagram of inclined shear mold; (**b**) Inclined shear specimen.

The shear strength between layers in the inclined shear test can be calculated by Equation (3):

$$\tau = \frac{\sqrt{2}P}{2s} \tag{3}$$

where $\tau$ is the shear stress at the interlayer interface, MPa; $P$ is the shear failure loading, N; $s$ is the interlayer contact area of the specimen, mm$^2$.

*3.3. Interlayer Pullout Test*

The pullout test directly assesses the bonding ability between the upper and lower layers of the pavement [32]. The direct pull-off test, which only considers the vertical tensile force, is the most commonly used method by researchers. However, the combined effect of horizontal force and tensile stress is highly likely to occur on the interlayer of

pavement under the comprehensive influence of vehicle load and ambient temperature fluctuation during practical service. Therefore, Liu et al. [33] studied the variation of interlayer bond strength of asphalt pavement under the influence of tack coat dosage, temperature, and horizontal push force based on a shear pull-off test with horizontal load (horizontal thrust). However, this test device has the disadvantages of complicated manufacturing and operation.

Consequently, the direct pullout test method was employed due to its convenience in test operation and the generalizability of the method. Figure 5 displays the pullout device utilized in the experiment. We also tested using a fast-loading rate of 50 mm per minute.

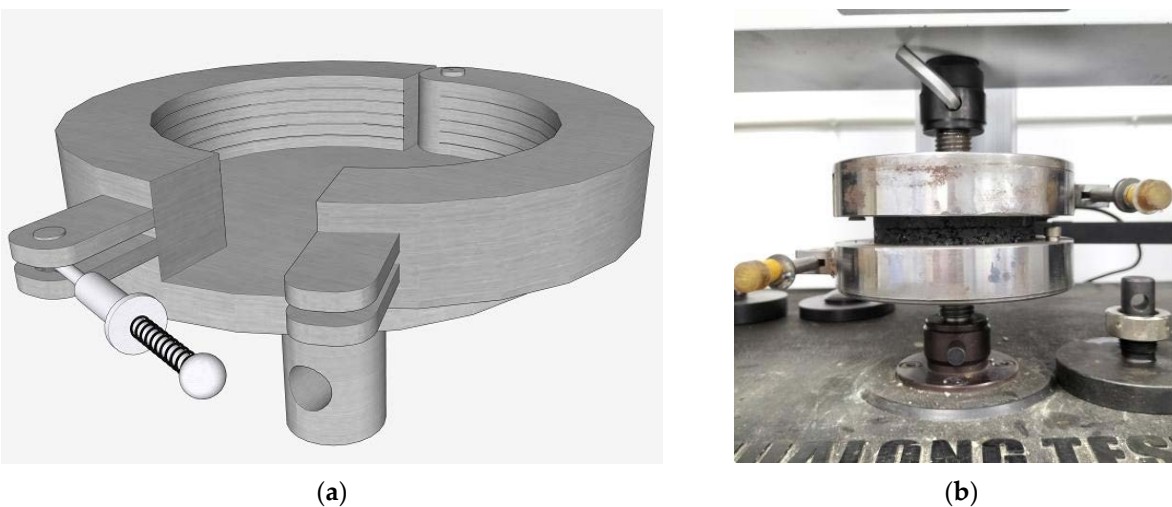

|        (**a**)        |        (**b**)        |

**Figure 5.** Pullout test: (**a**) Diagram of pullout device; (**b**) Pullout specimen.

The tensile strength between layers in the pullout test can be calculated by Equation (4):

$$C = \frac{F}{S} \tag{4}$$

where $C$ is the interlayer bonding strength, MPa; $F$ is the maximum tension value, kN; $S$ is the pullout area between layers, mm$^2$.

## 4. Results and Discussion

### 4.1. Analysis of Optimal Tack Coat Oil Volume

Three types of tack coat materials (matrix emulsified asphalt, SBS-modified emulsified asphalt, and SAE) were subjected to an inclined shear test at 25 °C. The sprinkling volumes of the three emulsified asphalts were 0.4 kg/m$^2$, 0.6 kg/m$^2$, 0.8 kg/m$^2$, 1.0 kg/m$^2$, and 1.2 kg/m$^2$, respectively. Finally, a polynomial curve was fitted to these test data. Finally, a polynomial curve was fitted to these test data. The results are shown in Figure 6.

Figure 6 illustrates the trend of shear strength values for the three types of emulsified asphalt. Within the scope of this research, as the amount of coating adhesive layer increases, the friction force initially increases and then decreases. The shear strength reaches its peak at a specific volume of tack coat oil application, indicating the optimal amount, after which the shear strength gradually decreases as the amount of tack coat oil increases. A comparison of the shear strength values among the three types of emulsified asphalt reveals that SAE demonstrates superior interlayer bonding performance compared to SBS-modified emulsified asphalt and matrix emulsified asphalt. By observing the trend of the curve changes in Figure 6, it can be found that when the dosage of the take coat material exceeds the optimum amount, the shear strength value shows a decreasing trend. Therefore, the dosage of the adhesive layer material needs to be set reasonably. Otherwise, the interlayer adhesion will be adversely affected, which could reduce the interlayer water resistance performance.

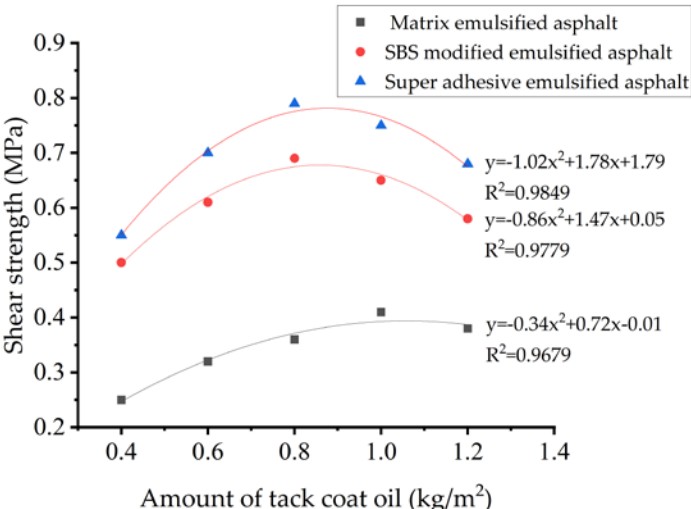

**Figure 6.** Effect of the amount of tack coat on the shear strength.

Moreover, the correlation coefficients of the three regression equations yielded squared $R^2$ values exceeding 95%, suggesting a solid alignment between the regression lines and the detection points. The result indicated a good fit of the equations. Consequently, the regression equation enables the calculation of the optimal sprinkling volumes for matrix emulsified asphalt, SBS-modified emulsified asphalt, and SAE, which are determined to be 1.1 kg/m², 0.85 kg/m², and 0.87 kg/m², respectively.

### 4.2. Effect of Different Adhesive Layer Materials on Interlayer Water Resistance

This study aimed to analyze the water resistance of SAE tack coat materials using matrix emulsified asphalt and SBS-modified emulsified asphalt as control groups. Following the dynamic water scouring test, the composite specimens were subjected to inclined shear and pullout tests at different temperatures (5 °C and 55 °C) to assess the interlayer water resistance of various tack coat materials by the ratio of the interlayer shear and tensile strength before and after scouring.

#### 4.2.1. Effect of Tack Coat Material on Interlayer Water Resistance at 5 °C

The results of shear strength and tensile strength tests for different tack coat materials before and after dynamic water scouring at 5 °C are presented in Figure 7.

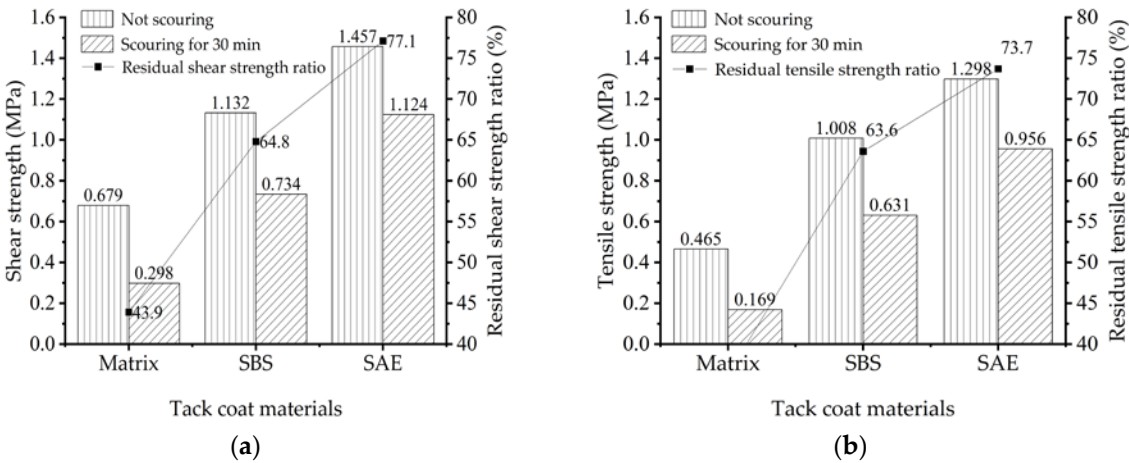

**Figure 7.** (**a**) Effect of different tack coat materials on the interlayer shear strength values after scouring; (**b**) Effect of different tack coat materials on the interlayer tensile strength values after scouring.

The change curves of residual shear and tensile strength ratios reveal noticeable variations in interlayer strength ratios for different tack coat materials under the same scouring time. Specifically, the strength value of the matrix emulsified asphalt has the maximum decline after scouring, indicating that when the pavement is in a low-temperature state, its water resistance is the weakest. SAE exhibits the least decrease in strength value after scouring, with an increase of 12.3% and 10.1% in residual shear and tensile strengths, respectively, compared to SBS-modified emulsified asphalt. Moreover, compared to matrix emulsified asphalt, SAE shows an increase of 33.2% and 37.4% in residual shear and tensile strengths, respectively, highlighting that it enhances interlayer water resistance when used as a binder oil under low-temperature conditions.

### 4.2.2. Effect of Tack Coat Material on Interlayer Water Resistance at 55 °C

The test results of shear strength and tensile strength of different tack coat materials before and after dynamic water scouring at 55 °C are shown in Figure 8.

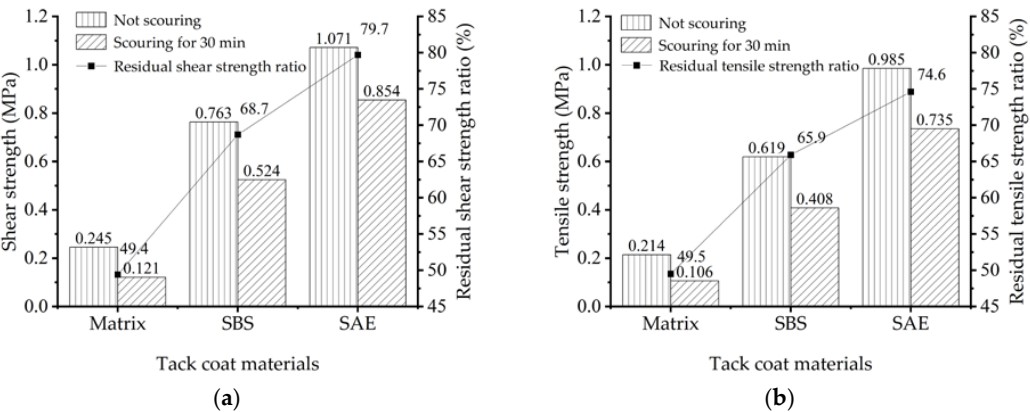

**Figure 8.** (**a**) Effect of different tack coat materials on interlayer shear strength values after scouring; (**b**) Effect of different tack coat materials on interlayer tensile strength values after scouring.

As shown in Figure 8, when the interlayer temperature is at a high temperature of 55 °C, the trends of the three tack coat materials before and after the dynamic water flushing are consistently compared with those at 5 °C. The curves of the residual tensile strength ratio demonstrate that SAE exhibits 11.0% and 8.7% higher residual shear and tensile strengths at high temperatures compared to SBS-modified emulsified asphalt and 30.3% and 25.1% higher compared to matrix emulsified asphalt. This finding suggests that SAE can significantly enhance interlayer water resistance at high temperatures when employed as a tack coat oil.

Comparison between the shear and tensile strength values of the composite specimens under the same temperature and scouring conditions reveals that the tensile strength values are generally lower than the shear strength values. This is because the force modes generated between the layers by the shearing and tensile test operations differ. The tensile test data only reflect the bonding performance of the emulsified asphalt adhesive layer material, while the shear test data contain not only the interlayer bonding force but also the frictional force between the aggregate particles of the composite specimens. Therefore, under the same conditions, the shear strength is slightly greater than the tensile strength.

As can be seen from the above temperature tests, given the most unfavorable temperature condition between layers, both in low- and high-temperature environments, the interlayer water resistance of SAE demonstrates a significant improvement compared to both SBS-modified emulsified asphalt and matrix emulsified asphalt.

### 4.3. Effect of Temperature on Interlayer Water Resistance

The influence of temperature on the water resistance performance of SAE was examined by analyzing the interlayer shear and tensile strength values before and after dynamic

water scouring. These values are utilized to assess the interlayer water resistance, as the experimental findings show in Figure 9.

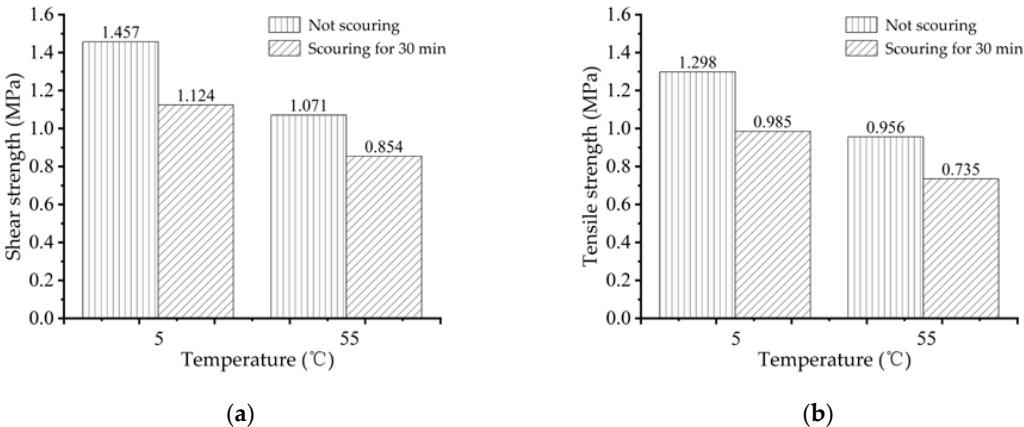

(a)          (b)

**Figure 9.** (**a**) Variations of interlayer shear strength at different temperatures; (**b**) Variations of interlayer tensile strength at different temperatures.

The interlayer shear and tensile strength's initial and final values exhibit significant variations at different temperatures, as illustrated in Figure 9. The composite specimens' shear and tensile strength values decrease with increasing temperature, both before and after dynamic water flushing, and the shear and tensile strengths at 55 °C are only about 75% of those at 5 °C. This can be attributed to the ability of the modified emulsified asphalt material to maintain its excellent performance in low-temperature conditions, reducing the likelihood of interlayer sliding under load. However, at the precondition of high temperatures and the same load, the emulsified asphalt material in the binder will undergo flow, leading to reduced bonding performance and increased susceptibility to interlayer slippage, with ultimate pavement breakage.

*4.4. Effect of Scouring Time on Interlayer Water Resistance*

The impact of dynamic water scouring time on the water resistance of SAE is examined by analyzing the interlayer shear and tensile strength and the percentage change in strength, with exhibition results shown in Figure 10.

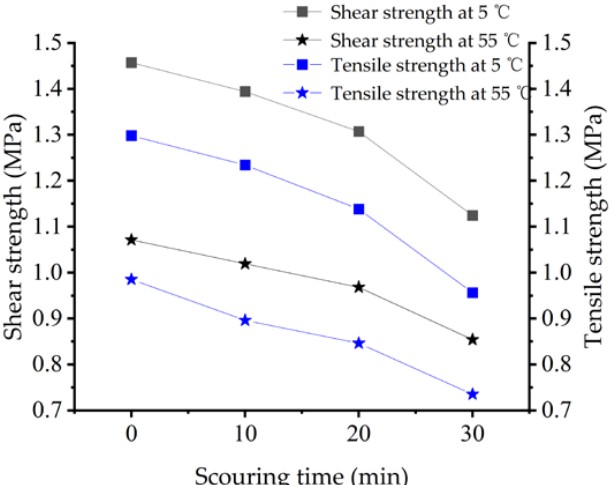

**Figure 10.** Interlayer shear and tensile strength are varied under the influence of different scouring times.

Figure 10 illustrates a decreasing trend in SAE's interlayer shear and tensile strength values with increasing scouring time at both low and high temperatures. Given the

consistent variation curves of shear and tensile strength values, we extracted the percentage decrease of shear strength concerning scouring time for analysis. The test results are presented in Table 4. The interlayer shear strength notably increases in its decreasing trend as the scouring time increases (Table 4). In the first 0–10 min, the shear strength decreases by approximately 5%, indicating a more substantial bonding effect between the emulsified asphalt and the tack coat during the initial stage of dynamic water scouring, resulting in enhanced scouring resistance. During the 10–20 min of the flushing experiment, there is no significant difference in the percentage decrease of shear strength compared to the initial stage. Within 20–30 min, the percentage decrease in shear strength exceeds 10%, suggesting the loss of tack coat oil during the later stage of dynamic water flushing, which leads to inadequate water resistance between the layers.

**Table 4.** Percentage decrease of shear strength with scouring time curve.

| Scouring Time/min | 5 °C | 55 °C |
|---|---|---|
| 0~10 | 4.32% | 4.86% |
| 10~20 | 6.24% | 5.00% |
| 20~30 | 14.00% | 11.78% |

## 5. Analysis of the Mechanism of Water Resistance of SAE

The evaporation residues of matrix emulsified asphalt, SBS-modified emulsified asphalt, and SAE were analyzed using Fourier infrared spectroscopy [34]. The samples were prepared using the coating method, and a Fourier-transform infrared spectrometer (FTIR spectrometer) was obtained with a resolution of 4 cm$^{-1}$, 32 scans, and a test range of 4000~400 cm$^{-1}$. The results of the analysis are presented in Figure 11.

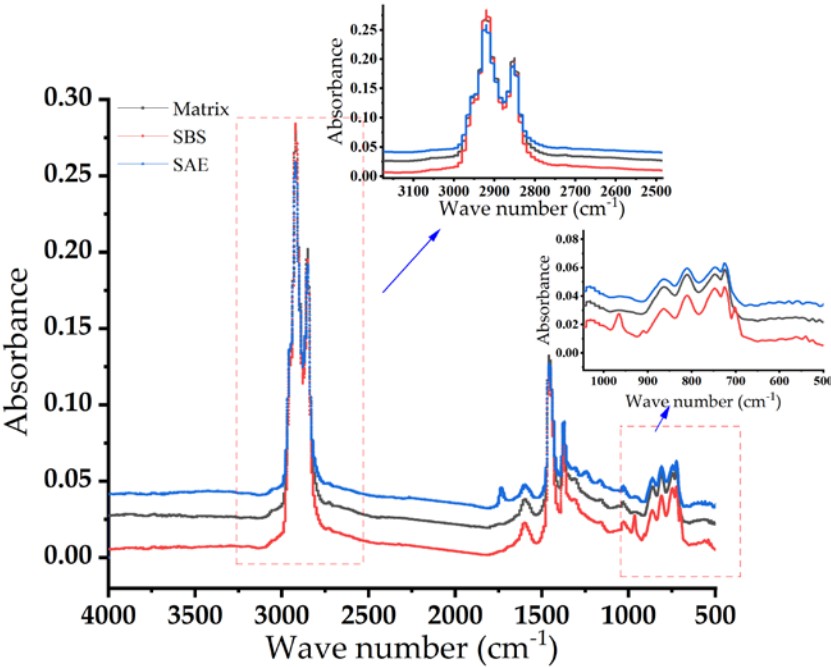

**Figure 11.** Infrared spectra of three kinds of emulsified asphalt.

Figure 11 illustrates that most peaks in the spectrum of the three emulsified asphalt samples exhibit similar positions within the test range, although their intensities show various peaks. A comparison between the evaporated residue spectra of matrix emulsified asphalt and SBS-modified emulsified asphalt reveals that the latter exhibits two additional absorption peaks at 965 cm$^{-1}$ and 700 cm$^{-1}$, corresponding to characteristic peaks of SBS. The absorption peak at 965 cm$^{-1}$ corresponds to the C-H bending vibration in the

butadiene-1,4 structure (-HC—CH-), a characteristic peak of the polybutadiene segment. The absorption peak at 700 cm$^{-1}$ is linked to the bending vibration of the substituted C-H on the benzene ring, indicating a characteristic peak of the polystyrene segment. The evaporated residue spectrum of 4% SBR and 12% PU-modified emulsified asphalt exhibits amplified (attenuated) absorption peaks compared to the evaporated residue spectrum of matrix emulsified asphalt, suggesting that the composite modifier interacted with specific components of the matrix asphalt, leading to alterations in its internal compositional structure. Therefore, the composite modification results from a combination of physical and chemical modifications, with the chemical modification primarily attributed to the PU modifier.

The PU molecular chain contains carbamates (-NHCOO-) and isocyanate-based (-NCO) groups, exhibiting high polarity and certain liveliness. Among them, isocyanates can undergo chemical reactions with phenol, anhydride, and carboxylic acid present in asphaltene [35], forming urea moiety and amide groups. The interactions among these groups enhance intermolecular forces, leading to increased cohesive energy and strengthened hydrogen bonding interactions between PU macromolecules. This has a significant effect on improving the bonding performance. Additionally, isocyanates can react with oxygen atoms, nitrogen atoms, and hydrogen-containing groups in asphalt to form hydrogen bonds. More hydrogen bonds increase the strength of van der Waals forces, leading to progressively stronger electrostatic forces between ions. The water-based polyurethane chosen for this experiment utilized water as its dispersion medium. The PU particles continuously approach and aggregate as water evaporates throughout the drying process, eventually forming a continuous and dense interfacial film.

The SBR modifier exhibits an asymmetric structure of polar and non-polar groups, in which the polar groups can enhance the molecularly oriented adsorption ability. Consequently, the material not only possesses strong cohesive forces but also demonstrates robust interfacial adsorption forces. The two modifiers exhibit synergistic effects, resulting in excellent bonding after being combined. Under dynamic water scouring, SAE exhibits superior adhesion to the aggregate surface compared to matrix emulsified asphalt and SBS-modified emulsified asphalt, enhancing its bonding performance, washout resistance, and water resistance.

## 6. Conclusions

This study examined the water resistance of SAE through the utilization of self-designed inclined shear, pullout, and dynamic water scouring devices. These devices were employed to measure SAE's shear and tensile strength under various conditions. The key findings of this research can be summarized as follows:

(1)  Within the adopted range of emulsified asphalt spraying volumes, the interlayer shear strength gradually increases with an augmented amount of tack coat material. However, an excessive quantity of tack coat material can reduce the interlayer shear strength. Through the application of curve fitting techniques to the shear strength values and spraying volume data, optimal spraying volumes for the interlayer of matrix emulsified asphalt, SBS-modified emulsified asphalt, and SAE at a temperature of 25 °C are determined to be 1.1 kg/m$^2$, 0.85 kg/m$^2$, and 0.87 kg/m$^2$, respectively.

(2)  SAE exhibits superior water resistance compared to SBS-modified emulsified asphalt and matrix emulsified asphalt at high and low temperatures. SAE's residual shear and tensile strengths at high temperature increase by 11.0% and 8.7% compared to SBS-modified emulsified asphalt and by 30.3% and 25.1% compared to matrix emulsified asphalt. The residual shear and tensile strengths at low temperatures increase by 12.3% and 10.1% compared to SBS-modified emulsified asphalt and by 33.2% and 37.4% compared to matrix emulsified asphalt.

(3)  After SBR and PU are mixed to form a composite latex, the emulsified asphalt system has physical and chemical modifications. Therefore, SBR and PU are suggested to

be used in combination to enhance the bonding performance, scouring, and SAE water resistance.

**Author Contributions:** Conceptualization, X.L.; methodology, X.L. and N.W.; data curation, Y.S. and N.W.; writing—original draft preparation, Y.S. and N.W.; writing—review and editing, X.L.; visualization, Y.S.; project administration, F.S.; funding acquisition, X.L., F.S. and B.P. All authors have read and agreed to the published version of the manuscript.

**Funding:** The authors appreciate the financial support by the National Natural Science Foundation of China (No. 51978401), the Key Laboratory of Performance Evolution and Control for Engineering Structures (Tongji University) (No. 2019KF-5), and the Open Fund of Shanghai Key Laboratory of Engineering Structure Safety (No. 2019-KF07).

**Institutional Review Board Statement:** Not applicable.

**Informed Consent Statement:** Not applicable.

**Data Availability Statement:** Not applicable.

**Conflicts of Interest:** The authors declare no conflict of interest.

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
