# Peer review of "Water Resistance of Super Adhesive Emulsified Asphalt Based on Dynamic Water Scouring"

_coatings, doi:10.3390/coatings13101776_

Round 1

Reviewer 1 Report

1.      Suggest title “Water Resistance of SAE Based on Dynamic Water Scouring”

2.      The main technical indicators of matrix asphalt, as depicted in Table 1, require further elaboration rather than just presenting raw data values.

3.      Need clarification regarding why SBR modifier latex was selected at levels of 2%, 4%, and 6%?

4.      The statement on page 3, line 92, indicating 500 revolutions per minute for 10 minutes appears to be excessively high. Do you have any supporting references for this?

5.      Provide the full names corresponding to the abbreviations AC-13 and AC-20.

6.      Explained clearly the specific formula from Figure 1. For instance, y=1.02x2+…; y=0.86x2+… and etc. In addition, what the correlation with R-square?

7.      Section 4.2, titled "Effect of adhesive layer material," lacks clarity and appears confusing. Is there a correlation or meaningful relationship that can be established in this context? It is methodology?

8.      Page 8, lines 225-227, requires additional discussion to provide a more comprehensive analysis.

9.      Table 4 title “percentage decrease” where the content is not reflected. For example, 4.32% to 4.86%, it is decreased?

Author Response

We feel great thanks for your professional review work on our article. As you are concerned, there are several problems that need to be addressed. According to your nice suggestions, we have made extensive corrections to our previous draft, the detailed corrections are listed below.

1. We think this is an excellent suggestion. We have changed the title to "Water Resistance of SAE Based on Dynamic Water Scouring".

2. The main specifications in Table 1 are the parameters of the matrix asphalt and do not require further elaboration.

3. We have added more references to support this idea (20).

4. According to the literature 21 and 22, it can be learnt that the mixing time is 1~10 min. Due to the actual mixing process in this test, the dispersion is not homogeneous when mixing for 5 min, so 10 min is chosen as the mixing time.

5. We have  provided the full names corresponding to the abbreviations AC-13 and AC-20, which are highlighted in red in the text.

6. The equation in Figure 1 is the fitted equation for the fitted curve, and R-squared is explained in the blue-font section of the text (lines 195-197).

7. Section 4.2 is intended to convey the effect of the use of different binder materials on the water resistance of the interlayer under complex conditions, highlighting the fact that SAE is more resistant to water than both base emulsified asphalt and SBS modified emulsified asphalt (lines 202-203 in blue in the text). The evaluation index is the ratio of interlayer shear and tensile strength before and after scouring (lines 206-207 in red in the text).

8.Lines 245-248 are a summary of the temperature tests above to emphasise that the water resistance of SAE is stronger than that of base emulsified asphalt and SBS modified emulsified asphalt.

9. I am very sorry that some of the data in Table 4 was not added due to an oversight on my part. Table 4 can be seen in correspondence with Fig. 10 and wants to express the degree of shear strength decay with increasing scour time.

Reviewer 2 Report

The article is interesting as is the topic of study. However, the authors must show a greater state of knowledge and discussion on the topic. Especially, show the novelty and originality of the study compared to other similar ones.

Author Response

We tried our best to improve the manuscript and made some changes to the manuscript. These changes will not influence the content and framework of the paper. And here we did not list the changes but marked in red in the revised paper. We appreciate for Reviewers' warm work earnestly and hope that the correction will meet with approval. 

Reviewer 3 Report

The research paper focuses on effect of dynamical water scouring on asphaly pavements as well as super adhesive emulsified (SAE) asphalt. The research is original and could be interesting for the readership of the Coatings. However, some clarification on methods used should be added.

There are some concerns that need to be addressed:

1)     Could you add some specification of the water used for the testing (elemental content, pH) since it could be varying a lot if we apply to the real condition?         

2)     Could you, please, justify why only temperature 5 and 55 C°was used for testing and not 20-25 C°, 0 C°and 60 C°for exemple?          

3)     What is the abrasion resistance of SAE and how often did it have to be replaced compared to regular used asphalt?

4)     It would be nice to have a brief comparison of how much the price of 1 km of asphalt pavement will increase if you use SAE asphalt.

5)     Have you ever considered the performance of the road build from the the mixture of plastic and bitumen?

Author Response

We feel great thanks for your professional review work on our article. As you are concerned, there are several problems that need to be addressed. According to your nice suggestions, we have made extensive corrections to our previous draft, the detailed corrections are listed below.

  1. The water used in the dynamic water souring test was ordinary domestic water, which was used to simulate the effect of dynamic water scouring on the tack coat after natural rainfall.(Lines 126-128 of the text in red)
  2. The basis for the temperature selection is in lines 142-148 of the text (marked in blue font) and is based on the actual temperatures at the project site and calculated by equations (1) and (2). The purpose is to know the water resistance of SAE in the worst temperature case.
  3. Thank you very much for your valuable comments, in response to questions 3 and 5 will be the next step of our research focus on the direction, will be reflected in the next paper.
  4.  Ordinary adhesive layer material used is base emulsified asphalt, the price of about 3000 yuan RMB a tonne, this paper uses SAE due to the addition of modifiers, the price of about 6000 yuan RMB a tonne. Spreading matrix emulsified asphalt needs the amount of tack coat oil is about 0.5kg/m2, spreading SAE the best amount of tack coat oil is 0.87kg/m2.

Round 2

Reviewer 1 Report

Accept in present form.

Author Response

We appreciate very much for your acceptance on our article. According to other reviewers' kind suggestions, we have made extensive corrections to our previous version of manuscript and the detailed corrections are shown in the paper. (All the revised contents are marked in green in the revised version of manuscript).

Reviewer 2 Report

I suggest improving the presentation of the images. For example, the font is very small. Additionally, better show the originality and innovation of the study in the introduction. Increase the discussion of the results by comparing them with similar studies.

Author Response

We appreciate very much for your professional review work on our article. As you are concerned, there are several problems that need to be addressed. According to your kind suggestions, we have made extensive corrections to our previous version of manuscript and the detailed corrections are listed below. (All the revised contents are marked in green in the revised version of manuscript).

  1. We have enlarged the font size in the images to better present these images.
  2. We have modified the Introduction to highlight the originality and innovation of the current study.
  3. Compared with the similar studies, the current research employes self-designed inclined shear, pullout, and dynamic water scouring devices to examine the shear and tensile strength of modified emulsified asphalt and SAE under various conditions of sprinkling volumes, temperatures, and scouring times. The results have been presented and compared with the previous similar studies.